# T cell Receptor Vβ9 in Method for Rapidly Quantifying Active Staphylococcal Enterotoxin Type-A without Live Animals

**DOI:** 10.3390/toxins11070399

**Published:** 2019-07-10

**Authors:** Reuven Rasooly, Paula Do, Xiaohua He, Bradley Hernlem

**Affiliations:** Western Regional Research Center, Foodborne Toxin Detection & Prevention Research Unit, Agricultural Research Service, United States Department of Agriculture, Albany, CA 94710, USA

**Keywords:** Staphylococcal enterotoxin type A, Superantigen: Raji B-cell, CCRF-CEM T-cell

## Abstract

Staphylococcal food poisoning is a result of ingestion of Staphylococcal enterotoxins (SEs) produced by *Staphylococcus aureus*. Staphylococcal enterotoxin type A (SEA) is the predominant toxin produced by *S. aureus* strains isolated from food-poisoning outbreak cases. For public safety, assays to detect and quantify SEA ideally respond only to the active form of the toxin and this usually means employing disfavored live animal testing which suffers also from poor reproducibility and sensitivity. We developed a cell-based assay for SEA quantification in which biologically-active SEA is presented by Raji B-cells to CCRF-CEM T-cells resulting in internalization of Vβ9 within 2 hours with dose dependency over a 6-log range of SEA concentrations. This bioassay can discern biologically active SEA from heat-inactivated SEA and is specific to SEA with no cross reactivity to the homologically-similar SED or SEE. In this study, we terminated any ongoing biochemical reactions in accessory cells while retaining the morphology of the antigenic sites by using paraformaldehyde fixation and challenged the current model for mechanism of action of the SEA superantigen. We demonstrated for the first time that although fixed, dead accessory cells, having no metabolic functions to process the SEA superantigen into short peptide fragments for display on their cell surface, can instead present intact SEA to induce T-cell activation which leads to cytokine production. However, the level of cytokine secretion induced by intact SEA was statistically significantly lower than with viable accessory cells, which have the ability to internalize and process the SEA superantigen.

## 1. Introduction

*Staphylococcus aureus* is a prevalent bacterial pathogen that produces a wide variety of exoproteins that cause various types of disease. Pathogenesis is mediated by virulence factors including some 23 different staphylococcal enterotoxins (SEs) that induce gastroenteric syndrome, exhibit emetic activity, and are the causative agents of food poisoning affecting 241,148 persons annually in the United States [1]. Some of these SE subtypes are active at very low concentration, as small as 1 fg/mL [2,3]. SEs function as superantigens that activate CD4+ T cells, cause proliferation of T-cells in a dose and time dependent manner [4], and induce differential regulation of CD154 [5] that is responsible for costimulatory signals to B cells. In addition, SEs induce differential expression of interferon-gamma (IFN-γ) [6], tumor necrosis factor (TNF) [7] and cytokine release in a dose-dependent manner [3]. Although superantigenic activity and the gastroenteric syndrome are two separate functions of SEs, there is a high correlation between these activities. The loss of emetic response has been shown to be correlated with the loss of T-cell activation [8,9]. When site-directed mutagenesis was used to inhibit SEC emetic activity, it also eliminated T-cell activation. Staphylococcal enterotoxin type A (SEA) is produced in larger quantities during the log phase of the bacterial growth cycle [10]. SEA is the most common etiological agent of the entire range of SEs encountered in food poisoning outbreaks [11,12]. It was shown that in the UK and in the US, SEA was the predominant SE, accounting for 78% of all toxin-producing *S. aureus* strains isolated from food-poisoning outbreaks cases [13,14]. Less than 200 ng of SEA can lead to disease [11,15,16]. The presently accepted methods to detect biologically-active SEA are bioassays that employ live animals such as monkeys or kittens with the induction of emesis as the observed response [17,18]. These costly, yet insensitive assays further suffer from poor reproducibility and are discouraged because of the ethical concerns regarding the use of live animals. To ensure food safety and to stop SEA from entering the human food chain while at the same time avoiding the use of live animals, new rapid detection assays for biologically-active SEA are needed. It has been shown that within 3 to 9 days, SEA induced the expansion of T-cell populations that bear TCR Vβ subsets 5.2, 5.3, 7.2, 9, 16, 18, and 22 in human T lymphocytes from PBMCs [19]. In this study, we examine the use of the level of the T-cell receptor (TCR) Vβ9 variant of the TCR β chain protein responsible for recognizing SEA in a human CD4+ T cell line for rapid detection of biologically-active SEA. The data presented in this study show, for the first time, that within 2 hours after stimulation with SEA, there is internalization of TCR Vβ9 as demonstrated by the reduction of TCR Vβ9 surface expression within a single T-cell line, and this phenomenon can be used for rapid detection and quantification of biologically-active SEA.

## 2. Results

### 2.1. Identify T-Cell Lines that Highly Express TCR Vβ9 Receptor on Their Surface

In search of a cell line that expresses TCR Vβ9 receptor on its surface that can be activated by SEA, we identified and examined three human T-cell lines. These cell lines were stained with phocoerythrin (PE) conjugated anti-human TCR Vβ9 antibody and analyzed by flow cytometry with a sample size of about 10,000 T-cells to identify which of them expressed the highest level of TCR Vβ9 protein on their surface. Figure 1 shows a histogram overlay of the results of the three labeled cell types compared with CCRF-CEM with no labeled antibody as an example of the baseline signal intensity. The geometric mean for each histogram for unlabeled CCRF-CEM and labeled SUPT1, MOLT3 and CCRF-CEM were 39.2, and 57.8, 93.1 and 905, respectively (left to right). Basically, these plots show that cells of the T lymphoblastoid line obtained from the peripheral blood of a 4-year-old Caucasian female with acute lymphoblastoid leukemia (CCRF-CEM) express the highest level of TCR Vβ9 protein on their surface. The mean intensity of TCR Vβ9 expression per CCRF-CEM cell was higher than the other cell types tested by 23, 15.6 and 9.7 times, respectively.

### 2.2. Evaluation of TCR Vβ9 Based Assay for Specific, Rapid Detection of SEA

SE subtypes share substantial amino acid sequence similarity. To evaluate if TCR Vβ9 can be used for specific rapid detection of biologically-active SEA and to assess the cross reactivity of the assay to other SE subtypes, CCRF-CEM T-cells in combination with Raji B cells were incubated for 2 hours with 1 µg/mL of SEA, SEB, SED or SEE. Flow cytometric data, illustrated in Figure 2, show that only SEA can mediate TCR Vβ9 internalization and substantially reduces TCR Vβ9 protein levels expressed on the surface of the CCRF-CEM T-cell line within the first 2 hours. The observed TCR Vβ9 internalization in this uniformly Vβ9 positive cell line should not be confused as contradicting the findings of Thomas et al. who report the expansion of T-cell populations expressing TCR Vβ9 in PBMCs 3 to 9 days after SEA stimulation. In this study, TCR Vβ9 internalization appears to be an early signal seen within 2 h after stimulation that precedes proliferation and rapid expansion of such Vβ9 positive cells as observed by Thomas et al. [19]. This result also shows that the TCR Vβ9-based assay is specific against SEA and does not exhibit cross reactivity with SEB, SED, or SEE. Stimulation with SEB, SED or SEE produced no response and did not decrease Vβ9 internalization even though SEA, SED and SEE belong to the same SE group [20,21] and share 70–90% sequence homology [22].

### 2.3. SEA Reduces TCR Vβ9 Protein Expression in a Dose Dependent Manner

To assess whether SEA will induce differential reduction of the TCR Vβ9 protein level, we incubated CCRF-CEM T-cells in combination with Raji B cells for 2 hours with increasing concentrations of SEA. Flow cytometric data, illustrated in Figure 3, shows that this in-vitro bioassay can discern biologically-active SEA from heat-inactivated SEA. Two hours after stimulation, SEA induces the reduction of TCR Vβ9 protein levels in a dose-dependent fashion over a 6-log range. This suggests that the TCR Vβ9-based assay can be used as an alternative to live animal methods which rely on the emetic response for quantification of biologically-active SEA, the form of the toxin that poses a threat to public health and safety.

### 2.4. Direct Activation of T Cell by PHA, But Not with SEA

Accessory cells are necessary for the reduction of TCR Vβ9 protein expression on CCRF-CEM T-cells using SEA but not using the plant lectin phytohemagglutinin (PHA). This is contrary to the findings of Wang et al. [23] in which it was reported that SEA alone, without any co-stimulatory molecule, could elicit a response from a T-cell clone in the absence of any accessory cells that process SEA into antigenic peptides and present the fragmented SEA components via their major histocompatibility complexes (MHCs) to the T-cell. Our results in Figure 4 stand in contrast to that finding and reveal that the presence of Raji B-cells bearing co-stimulatory molecules are essential to generate a secondary cellular signal required for this effect. The results show that without accessory cells there is no reduction in TCR Vβ9 protein level on CCRF-CEM T-cells exposed to SEA. On the other hand, following a 2-hour incubation with PHA, a substance found in abundance in raw legumes, T-cells are stimulated in a manner different than by SEA. PHA alone, without any co-stimulatory molecule and in the absence of accessory cells, interacts directly with CCRF-CEM T-cells to generate the primary cellular signal that induces internalization and substantial reduction of TCR Vβ9 protein levels within 2 h. As shown in Figure 4, stimulation by PHA was still as effective as by SEA. The reduction in TCR Vβ9 protein level by PHA was slightly greater than that produced by SEA with processing by Raji B cells which are shown to be efficient accessory cells for SEA.

### 2.5. Accessory Cells Are Essential for Activation of CCRF-CEM T-Cell With SEA

The importance of the Raji B cell line as an antigen-presenting cell (APC) is demonstrated by increased secretion of the signaling molecule interleukin 2 (IL-2) with a corresponding increase in the number of Raji B cells relative to T-cells. In the absence of Raji B cells, the SEA molecule is neither processed nor presented to the T-cells as by display on the MHC class II Raji B-cell motif, and without the presence of the co-stimulatory molecule on the Raji B cell line there is no secondary cellular signal produced, preventing its ability to stimulate T-cells and the secretion of IL-2 beyond background levels. Our results in Figure 5 shows that by adding 5000 Raji B-cells to 100,000 CCRF-CEM T-cell at a ratio of 1/20, the secretion of IL-2 increased by 6.7 times, measured by its OD450 from 0.0560 ± 0.002 to 0.3760 ± 0.0843. By adding 50,000 Raji cells to 100,000 CCRF-CEM T-cell at a ratio of 1:2, the secretion of IL-2 increased by 21.4 times to 1.1960 ± 0.0751 OD450.

### 2.6. SEA and PHA Stimulated Interleukin 10 (IL-10) Secretion by CCRF-CEM T-Cells is Accessory Cell Dependent

CCRF-CEM T-cells minimally require two cellular signals to become fully activated, leading to IL-10 production. The first cellular signal induces internalization and substantially reduces TCR Vβ9 protein expression level within 2 hours after stimulation. A second cellular signal induces action that stimulates CCRF-CEM T-cells to produce cytokines within 24 h after stimulation. Figure 6b shows that the presence of Raji B-cells is essential for full activation and production of IL-10 by CCRF-CEM T-cells. After stimulation for 24 h with PHA or SEA, secretion of IL-10 requires a second cellular signal from the co-stimulatory molecule on the accessory cells. In the absence of accessory cells, there is no IL-10 production. In addition, unlike SEA, PHA cannot induce the secretion of IL-2 in CCRF-CEM T-cells with or without Raji B-cells as accessory cells as shown in Figure 6a. These results contrast to the findings of Wang et al. [23] who reported that SEA alone in the absence of accessory cells could activate a T-cell clone.

### 2.7. T-Cells that Were Incubated with Fixed Accessory Cells Without any Metabolic Function Secreted Considerably Lower Levels of Cytokine

Antigenic processing and presentation are immunological mechanisms that internalize and degrade the antigen protein into small peptides. The fragmented peptides are presented by histocompatibility complexes (MHC) class II molecules on the surface of the antigen-presenting cell (APC) for recognition by T-cells. It is advantageous to use fixed dead cells where possible to reduce cell culture work. The current model for SEA activity suggests that dead accessory cells can be used as APC. Unlike regular antigens that require processing, SEA does not require processing to induce a T-cell response. Intact unfragmented SEA binds directly to the alpha-helical chain of the MHC class II, outside the peptide binding groove of APC [24,25,26,27] and also to the TCR β-chain to elicit T-cell response [28,29]. To challenge this hypothesis, Raji B cells were fixed with paraformaldehyde to render them unable to internalize and process antigens. To exam if Raji B cells fixed with paraformaldehyde retain cellular morphology of MHC class II molecules on their surface, which is important for presentation of intact antigen, fixed and unfixed Raji B cells were incubated with FITC-conjugated anti-MHC class II antibody. If fixation causes structural changes to the MHC class II molecule, it is expected that this will affect the binding of the labelled antibody to the MHC class II epitope. The flow cytometry results presented in Figure 7 show that paraformaldehyde fixation of the Raji B cells does not reduce the level of FITC-anti-MHC class II bound to the cells and it can be inferred that the cells retain their cellular morphology and MHC class II motifs intact. 

The results shown in Figure 8 demonstrate that the MHC class II on Raji B cells retains the ability to bind allophycocyanin-conjugated SEA when the cells are fixed before recognition by T-cells. 

The fixed Raji cells were incubated with the CCRF-CEM T-cell line and SEA. ELISA results in Figure 9 show that fixed Raji cells lacking any proteolytic processing can act as antigen-presenting cells, presenting intact SEA molecules to T-cells and stimulating the secretion of IL-2 and IL-10. Fixation destroys all metabolic function of the Raji cells and so they cannot actively internalize and proteolytically degrade SEA and display the short fragments of the degraded SEA on their cell surface, which is the typical mechanism of antigen presentation. The data suggest that this SEA detection assay does not require live accessory cells. However, T-cells that were incubated with fixed Raji B-cells secreted lower levels of IL-2 and IL-10 than T-cells that were incubated with unfixed live accessory cells. The result shows that there was a significant difference (*p* < 0.05) in the amount of secretion of IL-2 and IL-10 between treatment and control. By using IL-10, the limit of detection was 0.1 ng/mL of SEA for both fixed and unfixed live accessory cells. By using IL-2, the limit of detection was 1 ng/mL with unfixed live accessory cells and 10 ng/mL for fixed cells.

## 3. Discussion

The first methods developed for detection and quantification of active Staphylococcal enterotoxins, including SEA, relied upon the emetic response of live monkeys or kittens. Those assays were poorly reproducible and relatively insensitive. For many years, it has been desirable to find alternatives to live animal testing for detection of the activity of SEA, the predominant SE, accounting for 78% of all toxin-producing *S. aureus* strains isolated from food-poisoning outbreak cases [13,14]. The sensitive immunological assays and new recognition elements, such as aptamers that have been developed, are very sensitive, but they are incapable of discerning active SEA, which is a health hazard, from inactivated SEA which poses no threat to public health or safety. Even though gastroenteric syndrome and superantigenic activity are two distinct functions of SEs, there is a high correlation between these activities. It has been shown that structural alteration of the SEA molecule inhibits emetic activity and also inactivates its superantigenic activities [8,9]. It has been found that when site-directed mutagenesis was used to inhibit SEC emetic activity, it also eliminated T-cell activation. Therefore, it was proposed that measuring superantigenic activities that stimulate T-cell proliferation can predict emetic activities. Bavari et al. and Hufnagle et al. utilized human peripheral blood mononuclear cells (PBMC) and measured T-cell proliferation using 3H thymidine incorporation [30,31]. However, using human PBMCs adds complexity and they are not always obtainable. To overcome this obstacle, an ex-vivo bioassay, using splenocyte cells from sacrificed mice, can make detection processes more accessible for laboratory use. It was demonstrated that one mouse spleen can provide enough cells to replace approximately 500 live animal tests [4]. However, this ex-vivo bioassay involves the use of splenocyte cells from sacrificed mice and therefore raises the same ethical concerns about live animal testing. In the present study, we attempted to eliminate this concern by replacing the mouse splenocytes with a human cell line. 

While the mechanism of superantigenic activity is largely understood, there remains a recent dispute about the need for accessory cells to elicit a response from a T-cell clone by SEA. It is reported that SEA alone, without co-stimulatory molecule, can elicit a response from a T-cell clone in the absence of any accessory cells [23]. However, Kasper et al. disputed this finding and reported that MHC class II molecules play a significant role in the immune response to bacterial superantigens [32]. Unlike typical antigens, superantigens are not processed intracellularly. Instead, they bind to MHC class II molecules as intact macromolecules and bind outside of the peptide–antigen binding groove [24]. Superantigens also interact with the Vβ domains of TCRs, resulting in T cell stimulation [28]. Our goal in this paper was primarily to develop a new method for rapid detection (less than 2 hours) of SEA activity without relying upon live animals. In order to develop such a method, it is necessary to understand and confirm the mechanism of our cell-based technique. Our data shows that accessory cells are essential for activation of CCRF-CEM T-cell with SEA. In the absence of accessory cells and without co-stimulatory signal or processing, the level of IL-2 secretion was essentially at background levels. In this study, we terminated any ongoing biochemical reactions in Raji B cells by using paraformaldehyde fixation while retaining the morphology of the antigenic sites. The data shows that fixed, dead accessory cells, without any metabolic function and without any processing, are able to induce cytokine secretion. However, the level of cytokine secretion was substantially lower than with live accessory cells. Our data suggests that, contrary to current dogma, proteolytic processing and peptide presentation may be involved in generating a robust response to the superantigen SEA.

Our earlier work with the Jurkat T-cell line showed that TCR Vβ8 expression and IL-2 secretion can be used for the specific detection and quantification of biologically-active SEE [3,33]. The present study demonstrates, for the first time, that TCR Vβ9 expression can be used for rapid detection and quantification of biologically-active SEA. This response occurs within 2 hours of exposure to SEA. Subsequent secretion of cytokines IL-2 and IL-10 can be quantified by ELISA at 24 h. The level of cytokine secretion after stimulation with intact un-processed SEA, that reportedly binds outside of the peptide–antigen binding groove, was significantly lower than with processed SEA. This study shows that intact PHA molecules non-specifically reduce TCR Vβ9 protein levels expressed on the CCRF-CEM T-cells in the absence of accessory cells and without the safety mechanism requiring cellular interaction between two distinctive types of cells, accessory cell and T cell. However, it was shown that the mechanism for the activation of CCRF-CEM T-cells by intact SEA requires the presence and participation of accessory cells. While it may appear that the reduction of the level of TCR Vβ9 expression or internalization is contradictory with the reported expansion of T-cell populations expressing this and other TCR Vβs, it must be remembered that in this study we use a pure cell line and not PBMCs. Further, the latter expansion occurs much later, 3 to 9 days, compared with the 2 h response reported here. Although SEA, SED and SEE share considerable amino acid sequence similarity and consequently belong to the same group of SEs [20,21], this assay is specific to SEA with no cross reactivity to SED, SEE or SEB. Based on differential reduction of the TCR Vβ9 protein level, we can quantify SEA activity and discern between active and inactive forms of SEA. 

## 4. Materials and Methods

### 4.1. Materials and Reagents

SEs toxins SEA, SEB, SED, SEE, and biotinylated SEA were purchased from Toxin Technology (Sarasota, FL). SEs toxins were tested and confirmed as >95% pure by SDS-PAGE and Coomassie blue staining. SEs toxins solutions were prepared in water. Phytohaemagglutinin (PHA) was purchased from Sigma (St. Louis, MO, USA). RPMI (Roswell Park Memorial Institute medium) 1640, fetal calf serum (FCS), MEM non-essential amino acids, sodium pyruvate, and penicillin/streptomycin, were purchased from Gibco/Invitrogen (Carlsbad, CA, USA). PE-labeled mouse anti-human Vβ9 and PacBlue labeled mouse anti-human CD19 mAbs, as well as BD OptEIA ELISA kits for human IL-2 and human IL-10 were purchased from BD Biosciences (San Jose, CA, USA). Dead cell stain eFluor 780 was purchased from Thermo Fisher (Waltham, MA, USA).

### 4.2. Cells and Cell Culture

Cells of the CCRF-CEM (ATCC^®^ CCL-119™), MOLT-3 (ATCC^®^ CRL-1552™), and SUP-T1 [VB] (ATCC^®^ CRL-1942™) human lymphoblastoid T-cell lines and Raji (ATCC^®^ CCL-86™) human Burkitt’s lymphoma B cell line were purchased from the American Type Culture Collection (Rockville, MD). Cells were maintained in cell culture medium comprised of RPMI 1640 supplemented with 10% FCS, 1% MEM non-essential amino acids and 100 nM sodium pyruvate and containing 100 units/mL penicillin and 100 μg/mL streptomycin. Cell cultures were maintained under humidified atmosphere containing 5% CO_2_ in an incubator kept at 37 °C.

### 4.3. Assay Method

The incubation stage of the assay for active SEA was conducted using 96-well microplates in which the wells contained 50 μL of a 2 × 10^6^ cells/mL suspension of the respective line of T-cells in cell culture medium, 25 μL of a 2 × 10^6^ cells/mL suspension of Raji B-cells in cell culture medium and 25 μL of a sample of toxin or PHA at four-fold its final concentration. The cells were incubated for 2 h at 37 °C and then prepared for flow cytometric analysis as described below.

### 4.4. Quantitative Determinations of Active SEA by IL-2 or IL-10 Protein Secretion 

For determination of IL-2 and IL-10 cytokine secretion, cell suspensions of CCRF-CEM and Raji cells were prepared as above but incubated at 37 °C for 24 h. ELISA for IL-2 and IL-10 was performed on cell suspension supernatant following the kit manufacturer’s instructions. As we were not concerned with absolute cytokine concentrations, for relative comparison of secretion, the data for IL-2 and IL-10 levels were reported in terms of OD450. Negative control comprised a cell mixture with no added toxin or PHA. The SEA-containing samples were self-positive.

### 4.5. Flow Cytometry

Cells were prepared for flow cytometry by removing the media and washing the cells two times with PBS by spinning at 200× *g* for 10 min each. The cells were stained at 4 °C in the dark for 30 min and then washed two times to remove unbound stain. The cells were resuspended in PBS and flow cytometry was conducted on a BD Biosciences FACSAria Fusion cytometer (San Jose, CA, USA) equipped with violet (405 nm), blue (488 nm) and red (633 nm) lasers. Discernment of T and B cell populations was through staining with PacBlue-labeled anti-CD19 which binds to Raji B-cells. To exclude dead cells from analysis, they were stained with eFluor 780 dead cell stain. The relative level of TCR Vβ9 expression was measured by PE labeled anti-Vβ9 mAb. FlowJo software (FlowJo LLC, version 10.6.0, Ashland, OR, USA, 2019,) was used to analyze cytometric data sets. The cell population of interest was selected by forward and side light scatter to exclude debris and contaminants. Data are representative of three independent experiments.

### 4.6. Paraformaldehyde Fixation

Raji B-cells were fixed by incubation in PBS containing 1% paraformaldehyde at room temperature for 15 min. Fixed cells were washed three times in PBS and washed three more times in media before being used in the activation assays.

### 4.7. Statistical Analysis

Statistical analysis was performed on data sets using the Windows version of SigmaStat 3.5 from Systat Software (San Jose, CA, USA). Detection of SEA was analyzed by one-way ANOVA with at least three experimental repetitions. Statistical significance was determined with *p* < 0.05. A t-test of statistical significance was performed on treatments versus control.

## Figures and Tables

**Figure 1 toxins-11-00399-f001:**
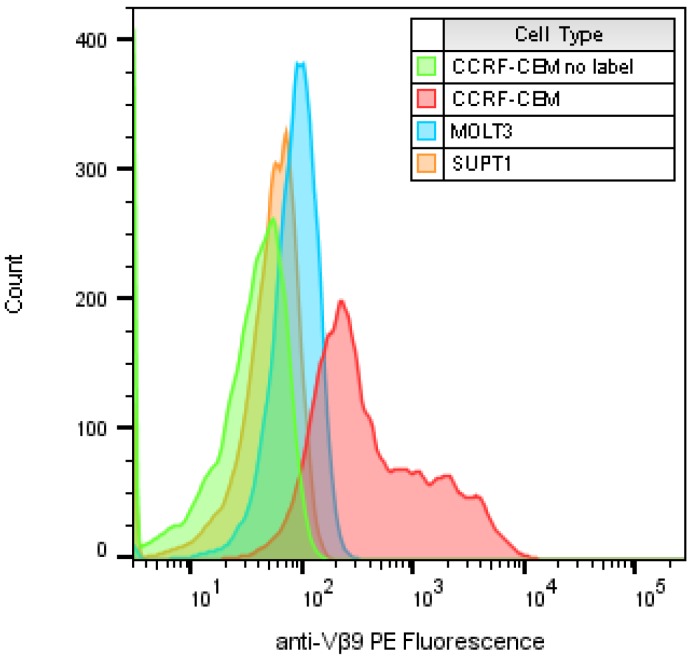
Flow cytometric analysis to compare the expression of TCR Vβ9 protein on the surface of three T-cell lines. All samples were labeled with PE-anti-Vβ9 except a “no label” CCRF-CEM control with no antibody. The x-axis represents the relative expression level of TCR Vβ9 protein per cell as relative fluorescence intensity on a logarithmic scale and the number of cells at each intensity displayed on the y-axis.

**Figure 2 toxins-11-00399-f002:**
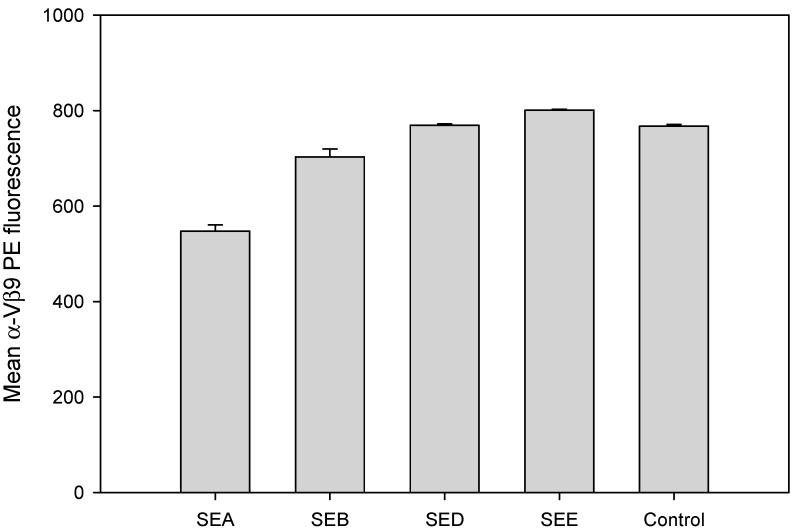
Assay specificity and cross-reactivity. SEA induced a specific reduction in TCR Vβ9 protein. CCRF-CEM T-cells in combination with Raji B cells were incubated for 2 hours with 1 µg/mL of SEA, SEB, SED or SEE. Cells were stained with PE-conjugated anti-Vβ9 monoclonal antibody (mAb) after stimulation. Fluorescence intensity was measured by flow cytometry. Data are representative of three independent experiments showing the mean florescence for the live T-cell population (about 8000 cells).

**Figure 3 toxins-11-00399-f003:**
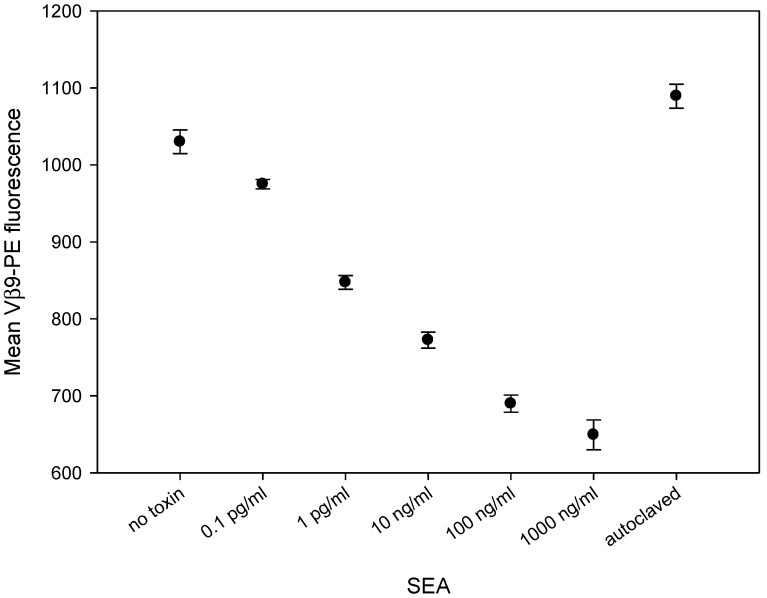
SEA induces reduction in TCR Vβ9 protein in a dose-dependent manner and can discriminate between biologically-active and inactive SEA. The mixed CCRF-CEM T-cells and Raji B cells were co-incubated with increasing concentrations of SEA for 2 h. After stimulation, the cells were stained with PE-conjugated anti-Vβ9 mAb. Mean fluorescence intensity for Vβ9 in live T-cell population was measured by flow cytometry. Data are representative of three independent experiments.

**Figure 4 toxins-11-00399-f004:**
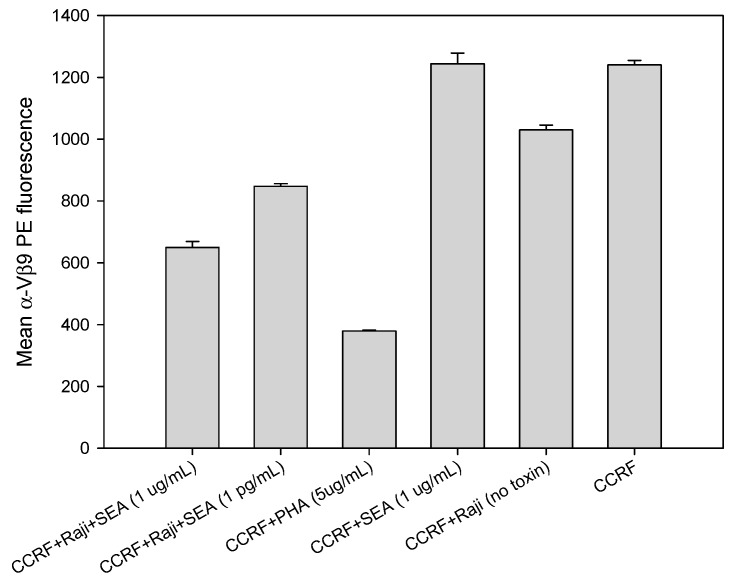
Raji B cells are essential for activation of T cell with SEA but not with the lectin PHA. CCRF-CEM T-cell line was plated with or without Raji B cells in a 96-well plate and incubated for 2 h with SEA at concentrations of 1 μg/mL and 1 pg/mL or with PHA at a concentration of 5 μg/mL.

**Figure 5 toxins-11-00399-f005:**
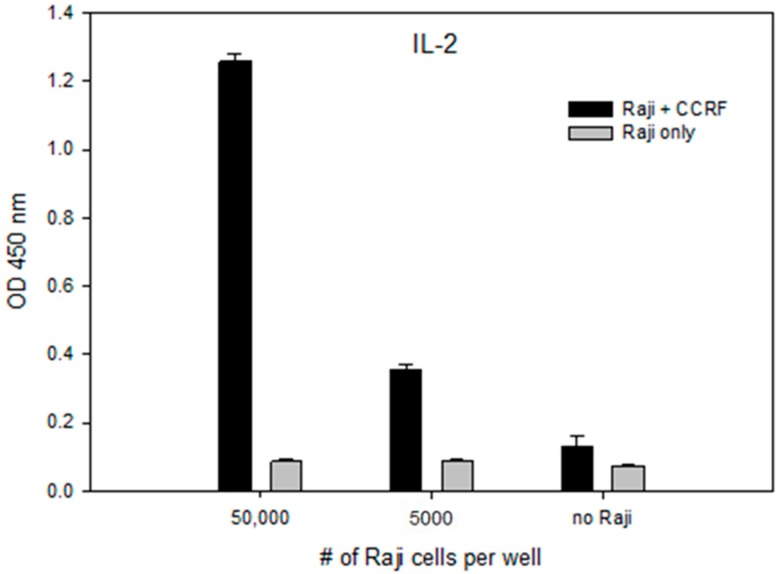
Raji B cells are required for stimulation of T cells with SEA. CCRF-CEM T-cells were plated with, without or with a reduced number of Raji cells in a 96-well plate and incubated for 24 h with SEA at 10 ng/mL concentration. IL-2 secretion measured by ELISA. Error bars represent standard errors.

**Figure 6 toxins-11-00399-f006:**
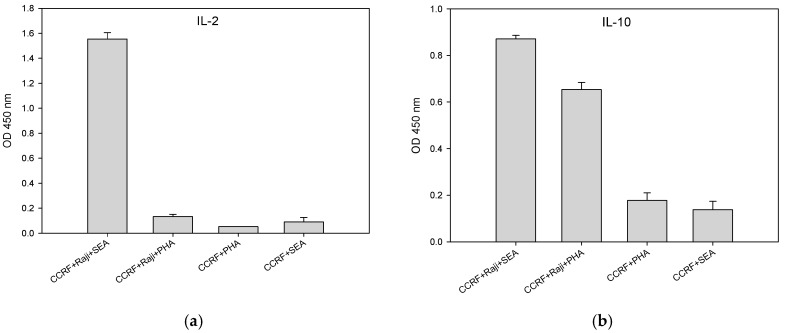
Production of IL-10 is accessory cell-dependent. CCRF-CEM T-cells and Raji B-cells were mixed in co-culture and were stimulated for 24 h with SEA at concentrations of 10 ng/mL or with PHA at concentration of 5 μg/mL. Induced IL-2 (**a**) and IL-10 (**b**) secretion was measured by ELISA. Error bars represent standard errors.

**Figure 7 toxins-11-00399-f007:**
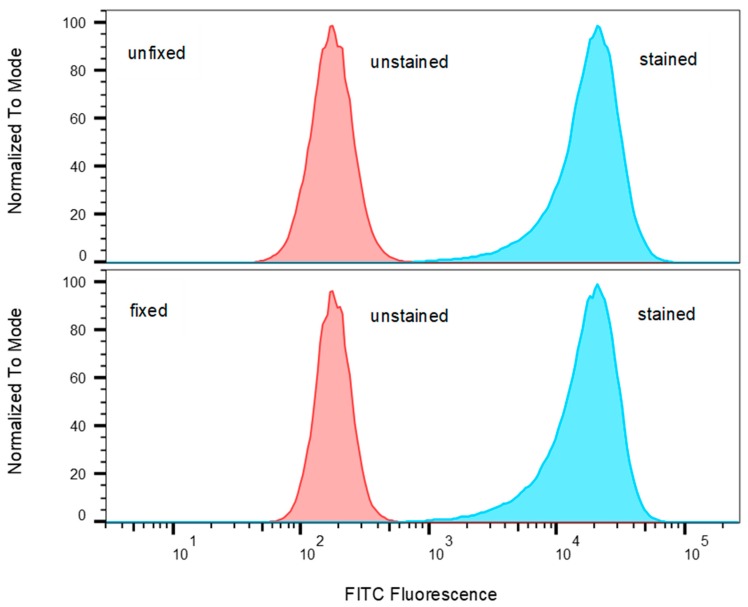
Comparison of Raji B cells stained with FITC-anti-MHC class II against unstained cells both unfixed and fixed. Histograms of FITC fluorescence were normalized to the modal value.

**Figure 8 toxins-11-00399-f008:**
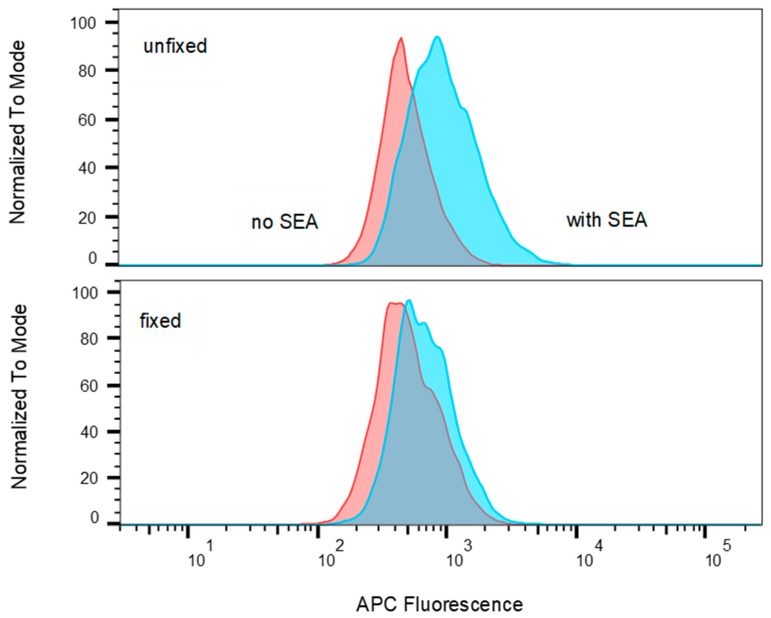
Comparison of allophycocyanin (APC) fluorescence in Raji B cells incubated with and without APC conjugated SEA where cells were either unfixed or fixed. Histograms have been normalized to the modal values.

**Figure 9 toxins-11-00399-f009:**
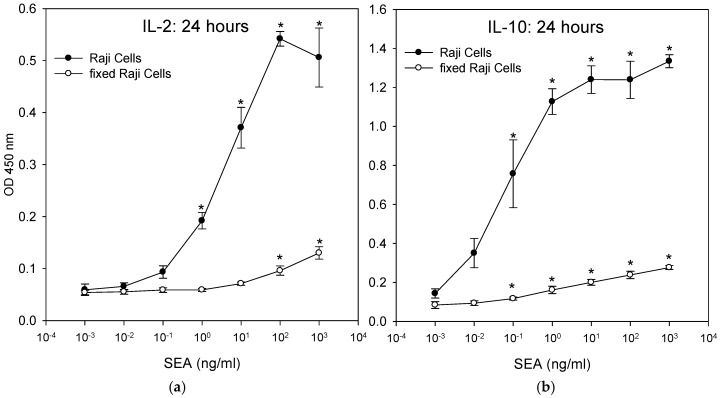
Activation of T-cells by processed SEA superantigen leads to substantially higher cytokine production. CCRF-CEM T-cells and metabolically-functioning Raji B-cells or fixed dead Raji B-cells were mixed in co-culture and were incubated for 24 h with increasing concentrations of SEA. Induction of IL-2 (**a**) and IL-10 (**b**) secretion was measured by ELISA. Error bars represent standard errors.

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
