# Peer review of "T cell Receptor Vβ9 in Method for Rapidly Quantifying Active Staphylococcal Enterotoxin Type-A without Live Animals"

_toxins, 2019, doi:10.3390/toxins11070399_

Round 1

Reviewer 1 Report

The manuscript addresses the rapidly SEA-detecting method using T cell line which express TCR Vbeta9. And authors appeal the method is useful for reduction of live animals.

The authors employ the ELISA for detecting IL-2 and IL-10. Then, ELISA for detecting SEA is more direct and faster. This should be commented in the Discussion.

In order to detect superantigenic activity of SEs, cytokine production assay using the other cells generally have been used. This should be commented in the Discussion.

Line 41

"Although superantigenic activity and the gastroenteric syndrome are two separate functions of SEs, there is a high correlation between these activities. "

Line 79-82

"To evaluate if TCR Vβ9 can be used for specific rapid detection of biologically active SEA"

The experiment design is not appropriate. SEA is not the only SE that binds to TCR Vbeta9. Because SEI, SEM, SEN, SEO and SES also bind to TCR Vbeta9, these toxins are required instead of SEB-SEE in the experiment.

In addition, why the specificity is required in the experiment?

To detect SEs-contaminated food in many of samples, the method should be able to detect many types of SEs at low cost.

2.3

The authors show heat inactivated SEA is equal to no toxin. However, there is no evidence that this toxin has lost emetic activity. If the undetectable heat-treated SEA has emetic activity, it is more dangerous for food that is safe to cause food poisoning.

Figure 4

CCRF+Raji (SEA 0) is requied.

Confirm the value of concentration (1 pg/ml or 1 ng/ml?)

2.7 and Discussion

The mechanism of superantigenic activity is well known. The authors should be shortened the description about the mechanism to become more concise.

Author Response

Reviewer #1

The manuscript addresses the rapidly SEA-detecting method using T cell line which express TCR Vbeta9. And authors appeal the method is useful for reduction of live animals.

 The authors employ the ELISA for detecting IL-2 and IL-10. Then, ELISA for detecting SEA is more direct and faster. This should be commented in the Discussion.

As suggested by the reviewer we have added this subject to the Discussion.

In order to detect superantigenic activity of SEs, cytokine production assay using the other cells generally have been used. This should be commented in the Discussion.

 As suggested by the reviewer we have added this subject to the Discussion that other cell line such as the Jurkat cell line was used for detection of SEE and we have cited an appropriate reference.

Line 41

"Although superantigenic activity and the gastroenteric syndrome are two separate functions of SEs, there is a high correlation between these activities. "

Other researchers have found that when site-directed mutagenesis was used to inhibited SEC emetic activity it also eliminated T-cell activation.

Line 79-82

"To evaluate if TCR Vβ9 can be used for specific rapid detection of biologically active SEA"

The experiment design is not appropriate. SEA is not the only SE that binds to TCR Vbeta9. Because SEI, SEM, SEN, SEO and SES also bind to TCR Vbeta9, these toxins are required instead of SEB-SEE in the experiment.

We agree with the reviewer that SEA is not the only toxin subtype that binds to TCR Vβ9, however, at present there are no commercial sources for the toxin subtypes cited by the reviewer. Further, these toxins are not known to be important foodborne contaminants where SEA is the predominant SE, accounting for 78% of all toxin producing S. aureus strains isolated from food poisoning outbreak cases [13,14].

In addition, why the specificity is required in the experiment?

It has been shown by others that the therapeutic application of antibodies which neutralize the biological effects of SEs have shown clinical success so, in order to neutralize the activity of a specific SE it is important to identify which SE is present.

To detect SEs-contaminated food in many of samples, the method should be able to detect many types of SEs at low cost.

We agree with the reviewer that it is desirable to detect all SEs in one activity assay, however, as yet there have not been identified cell lines that respond to all SE subtypes. Even though SEA and SEE belong to the same SE group and share 90% amino acid sequence homology, our results show that this TCR Vβ9 based assay is specific to SEA and cannot be used for detection of SEE, SED or SEB.

2.3

The authors show heat inactivated SEA is equal to no toxin. However, there is no evidence that this toxin has lost emetic activity. If the undetectable heat-treated SEA has emetic activity, it is more dangerous for food that is safe to cause food poisoning.

It has been shown by others that emetic activity and superantigenic activity are related.  When site-directed mutagenesis was used to inhibited SEC emetic activity it also eliminated T-cell activation [8,9]. By this logic, it is reasonable to understand that heat inactivation which inhibits T-cell activation will also inhibit emetic response. Moreover, the use of live animals to confirm this would be ethically disfavored and require requisite facilities which we do not have.

Figure 4

CCRF+Raji (SEA 0) is requied.

This additional control has been added to the figure.

Confirm the value of concentration (1 pg/ml or 1 ng/ml?)

The correct concentration is noted in the figure but was incorrectly noted in the legend. We thank the reviewer for catching this error which has been corrected. 

2.7 and Discussion

The mechanism of superantigenic activity is well known. The authors should be shortened the description about the mechanism to become more concise.

The reason that we discuss this in relative detail is that there remains recent dispute about the need for accessory cells to elicit a response from a T-cell clone by SEA. For example, please see the recent reference 22 where it was reported that SEA alone, without co-stimulatory molecule, could elicit a response from a T-cell clone in the absence of any accessory cells. Moreover, our results demonstrate for the first time that we actually could show internalization of Vβ9 in the absence of any accessory cells and without the safety mechanism requiring cellular interaction between accessory cell and T cell but only with the plant lectin phytohemagglutinin (PHA) and not with SEA. In addition, we are not aware of any study that shows internalization of TCR Vβ9 with or without accessory cells. This is the first study that demonstrates that TCR Vβ9 expression on the CCRF-CEM T-cell line combined with Raji B cells can be used for quantifying SEA. This study also suggests that the SEA detection assay does not require live accessory cells. Fixed Raji B cells lacking any proteolytic processing can act as antigen-presenting cells, presenting intact SEA molecules to T-cells and stimulating secretion of IL-2 and IL-10. However, T-cells that were incubated with fixed APCs lacking any proteolytic processing elicit substantially lower T-cell response as demonstrated by secreted lower levels of IL-2 and IL-10 than T-cells that were incubated with unfixed live accessory cells that have metabolic functions that can degrade SEA and display short fragments of the degraded SEA on their cell surface.

Reviewer 2 Report

This manuscript reports a flow cytometry-based bioassay for the detection of staphylococcal superantigens and suggests that, contrary to current dogma, proteolytic processing and peptide presentation may be involved in generating a robust response to superantigen A. Overall, this paper is lacking in appropriate controls, replicate data and sufficient data to support their claims. 

Major Comments:

1.    The major goal of the paper is unclear, as several goals are suggested. 

2.    If the goal is to develop a new bioassay, it is crucial to consider the reproducibility and ability to standardize the assay.  Specifically, it should be objective and quantitative, for instance measuring IL-2 release using a purified standard for IL-2. It is not clear how TCR Vb8 expression on T cells can be standardized between assays, users, and cytometers. The new assay should be compared against the currently accepted assay. 

3.    If the goal is to show a new and unappreciated mechanism by which superantigens can activate T cells, then additional work needs to be performed to demonstrate this mechanism. A considerable body of work supports the currently accepted mechanism, including the ability of soluble TCR Vb protein to block activation. Thus the authors would need ot discuss these data I the context of their data. Also, the counter argument to the authors claims is that it is very likely that the paraformaldehyde fixation of the Raji B cells damages the structure of the class II MHC and thus the superantigen is no longer able to bind it. Thus, experiments to show that MHC class II antibodies and superantigen still bind to the fixed cells would be very important. 

4.    Error bars should be provided and the number of replicates shown and type of error bars shown for all figures. For example, in Figure 2, the reader cannot conclude that SEA has lower TCR expression that the other if no error bars are shown. 

Minor comments

5.    Figure 1 – why are there two peaks for the cell line CCRF-CEM? What is the “no label” control with this cell line – the appropriate control would be no primary antibody to determine whether the smaller peak is due to non-specific binding or perhaps cellular Fc receptors binding to the primary antibody. 

6.    Is detection of inactive SEA really a problem? (line 202, motivation for the need for a new assay)

7.    The discussion should be broken into several paragraphs.

8.    The paper would benefit from editing for English.

Author Response

Reviewer #2

This manuscript reports a flow cytometry-based bioassay for the detection of staphylococcal superantigens and suggests that, contrary to current dogma, proteolytic processing and peptide presentation may be involved in generating a robust response to superantigen A. Overall, this paper is lacking in appropriate controls, replicate data and sufficient data to support their claims. 

 As suggested by the reviewer, we have added appropriate controls and replicate data.

 Major Comments:

The major goal of the paper is unclear, as several goals are suggested. 

Our goal in this paper was primarily to develop a new method for rapid detection (less than two hours) of SEA activity without relying upon live animals. In this study we show for the first time the successful implementation of Vβ9 for rapid quantification of active SEA as an alternative to live animal testing. In order to develop such a method and reduce cell culture work, it is necessary to understand and confirm the mechanism of our cell-based technique. It is advantageous to use fixed dead cells where possible to reduce cell culture work. To test if dead cells can replace live cell we used paraformaldehyde fixed dead Raji B cells that retain their cellular morphology. Our results show that live accessory cells are not required for presenting SEA. Fixed Raji B cells lacking any proteolytic processing and that cannot internalize the toxin or conduct any proteolytically degradation or display the toxin short fragments on their cell surface actually can act as antigen-presenting cells, presenting intact SEA molecules to T-cells and stimulating secretion of IL-2 and IL-10. However, T-cells that were incubated with fixed Raji B-cells had lower T-cell activation and thus secreted lower levels of IL-2 and IL-10 than T-cells that were incubated with unfixed live accessory cells.  

If the goal is to develop a new bioassay, it is crucial to consider the reproducibility and ability to standardize the assay.  Specifically, it should be objective and quantitative, for instance measuring IL-2 release using a purified standard for IL-2. It is not clear how TCR Vb8 expression on T cells can be standardized between assays, users, and cytometers. The new assay should be compared against the currently accepted assay. 

We previously reported representative results from triplicate experiments using 10,000 cells per sample. We have reworked the figures and added error bars to show the statistical reproducibility. The currently accepted assays for SEA activity are the monkey and kitten assays which, as we noted previously, raise ethical concern. IL-2 is unstable and it is difficult to rely upon absolute values of IL-2 secretion. We were not concerned with absolute cytokine concentrations, for relative comparison of secretion, the data for IL-2 and IL-10 levels were reported in terms of OD450. For this reason, and as noted for TCR Vβ9 expression, it is sufficient to rely upon relative or normalized response for comparison of assay results.

If the goal is to show a new and unappreciated mechanism by which superantigens can activate T cells, then additional work needs to be performed to demonstrate this mechanism. A considerable body of work supports the currently accepted mechanism, including the ability of soluble TCR Vb protein to block activation. Thus the authors would need ot discuss these data I the context of their data. Also, the counter argument to the authors claims is that it is very likely that the paraformaldehyde fixation of the Raji B cells damages the structure of the class II MHC and thus the superantigen is no longer able to bind it. Thus, experiments to show that MHC class II antibodies and superantigen still bind to the fixed cells would be very important. 

Our results show (Figure 7) that fixed, dead cells that retained their cellular morphology but have no metabolic function, lacking any proteolytic processing and that cannot internalize the toxin or conduct any proteolytically degradation or display the toxin short fragments on their cell surface actually can act as antigen-presenting cells, presenting intact SEA molecules to T-cells and stimulating secretion of IL-2 and IL-10 in dose dependent manner. However, live B-cells that have metabolic activity and are able to process SEA produce a much higher cytokine response in T-cells.

Error bars should be provided and the number of replicates shown and type of error bars shown for all figures. For example, in Figure 2, the reader cannot conclude that SEA has lower TCR expression that the other if no error bars are shown. 

As suggested by the reviewer, error bars and replicate information was added to all figures.

Minor comments

5.    Figure 1 – why are there two peaks for the cell line CCRF-CEM? What is the “no label” control with this cell line – the appropriate control would be no primary antibody to determine whether the smaller peak is due to non-specific binding or perhaps cellular Fc receptors binding to the primary antibody. 

All samples were stained with PE-anti Vβ9 except the one identified as “no label”. This is the appropriate control that the reviewer references. The label was on the primary antibody and no unlabeled primary antibody was used. The cause of the second peak is unknown but consistent and reproducible. [text was edited to clarify these points]

6.    Is detection of inactive SEA really a problem? (line 202, motivation for the need for a new assay)

There are situations where inactivated toxin may be present as when testing processing technologies and treatments for the elimination of toxin from contaminated food or for the prevention of toxin transmission through food. In such cases it is vital to be able discern active SEA, which poses a threat to public health and safety, from inactivated SEA.

7.    The discussion should be broken into several paragraphs.

As suggested by the reviewer, the Discussion has been edited.

8.    The paper would benefit from editing for English.

As suggested by the reviewer, the English was edited.

Reviewer 3 Report

1) The authors do not mention the developing bioassays based on new recognition elements, such as aptamers for SEA detection, which do not involve animals. A comment should be added in Discussion (Wu, S., Duan, N., Gu, H., Hao, L., Ye, H., Gong, W., & Wang, Z. (2016). A Review of the Methods for Detection of Staphylococcus aureus Enterotoxins. Toxins, 8(7), 176. doi:10.3390/toxins8070176)

2) The affinity and specificity of the beta chain of the T-cell receptor for SEA detection should be mentioned in Introduction (Sharma P, Postel S, Sundberg EJ, Kranz DM. (2013). Characterization of the Staphylococcal enterotoxin A: Vβ receptor interaction using human receptor fragments engineered for high affinity. Protein Eng Des Sel., 26(12), 781-9. doi: 10.1093/protein/gzt054.)

3) The word quantification should be added in the title

Author Response

Reviewer #3

The authors do not mention the developing bioassays based on new recognition elements, such as aptamers for SEA detection, which do not involve animals. A comment should be added in Discussion (Wu, S., Duan, N., Gu, H., Hao, L., Ye, H., Gong, W., & Wang, Z. (2016). A Review of the Methods for Detection of Staphylococcus aureus Enterotoxins. Toxins, 8(7), 176. doi:10.3390/toxins8070176)

As suggested by the reviewer, we have added additional discussion on this point.

The affinity and specificity of the beta chain of the T-cell receptor for SEA detection should be mentioned in Introduction (Sharma P, Postel S, Sundberg EJ, Kranz DM. (2013). Characterization of the Staphylococcal enterotoxin A: Vβ receptor interaction using human receptor fragments engineered for high affinity. Protein Eng Des Sel., 26(12), 781-9. doi: 10.1093/protein/gzt054.)

As suggested by the reviewer, we have added additional discussion in the Introduction on this point.

The word quantification should be added in the title

Suggestions adopted, we have edited the Title to include “Quantification”.

Reviewer 4 Report

In contrast to preview study measuring Vbeta T cells expansions, this study measure a decrease of Vbeta 9 expression on T cell to quantitatively detect SEA.

Minor comments

If measured, authors could be add some data about Vbeta 1, 2, 12 and 14 expressions in their model. These Vbeta repertoires are usually largely expanded after SEA stimulation. Some data could be interesting

To use their model in “routine practice”, Authors could added some data how to prepare vomiting or food samples to detect and quantify SEA by measuring the Vbeta 9 decrease.

To enlarge the discussion, authors could add comment about the SEB, SED, SEE, etc. detection in their in vitro model: use of other Vbeta target, etc.

Author Response

Reviewer #4

In contrast to preview study measuring Vbeta T cells expansions, this study measure a decrease of Vbeta 9 expression on T cell to quantitatively detect SEA.

Our study measured a decrease of TCR Vβ9 expression within two hours after stimulation with SEA and we applied this behavior for the quantitative detection of active SEA.

Minor comments

If measured, authors could be add some data about Vbeta 1, 2, 12 and 14 expressions in their model. These Vbeta repertoires are usually largely expanded after SEA stimulation. Some data could be interesting

In this report we are working with a specific T-cell line that is clonally pure. If we were to use splenocytes, PBMCs or such, the reviewer’s comments are correct and relevant but, in our application, the CCRF-CEM cell line expresses only Vβ9 and will not express these other repertoires even if stimulated with SEA.

To use their model in “routine practice”, Authors could added some data how to prepare vomiting or food samples to detect and quantify SEA by measuring the Vbeta 9 decrease.

In this report we demonstrate proof of concept. However, we have previously reported methods that used immunomagnetic beads that were coated with an anti-SEA antibody to eliminate food matrix interference and increased the signal-to-noise ratio and those techniques can be applied likewise in this instance.

To enlarge the discussion, authors could add comment about the SEB, SED, SEE, etc. detection in their in vitro model: use of other Vbeta target, etc.

As suggested by the reviewer, we have added additional discussion for other Vβ target.

Previse study shown that within 48h SEA induce the expansion of TCR Vβ subsets 5.2, 5.3, 7.2, 9, 16, 18 and 22 in human T lymphocytes from PBMC.  Thomas, D.; Dauwalder, O.; Brun, V.; Badiou, C.; Ferry, T.; Etienne, J.; Vandenesch, F.; Lina, G. Staphylococcus aureus superantigens elicit redundant and extensive human Vbeta patterns. Infect. Immun. 2009, 77, 2043–2050.

We thank the reviewers for their constructive comments.

Sincerely,